METHODS

# *k*Mermaid: Ultrafast metagenomic read assignment to protein clusters by hashing of amino acid *k*-mer frequencies

Anastasia Lucas[1,2], Daniel E. Schäffer[2,3], Jayamanna Wickramasinghe[4], Noam Auslander[2,5]*

1 Genomics and Computational Biology Graduate Group, University of Pennsylvania - Perelman School of Medicine, Philadelphia, Pennsylvania, United States of America, 2 Molecular and Cellular Oncogenesis Program, Ellen and Ronald Caplan Cancer Center, The Wistar Institute, Philadelphia, Pennsylvania, United States of America, 3 Computer Science and Artificial Intelligence Laboratory & Computational and Systems Biology Program, Massachusetts Institute of Technology, Cambridge, Massachusetts, United States of America, 4 Bioinformatics Facility, The Wistar Institute, Philadelphia, Pennsylvania, United States of America, 5 Department of Cancer Biology, University of Pennsylvania, Philadelphia, Pennsylvania, United States of America

* nauslander@wistar.org

## Abstract

Shotgun metagenomic sequencing can determine both the taxonomic and functional content of microbiomes. However, functional classification for metagenomic reads remains highly challenging as protein mapping tools require substantial computational resources and yield ambiguous classifications when short reads map to homologous proteins originating from different bacteria. Here we introduce *k*Mermaid for the purpose of uniquely mapping bacterial short reads to taxa-agnostic clusters of homologous proteins, which can then be used for downstream analysis tasks such as read quantification and pathway or global functional analysis. Using a nested hash map containing amino acid *k*-mer profiles as a model for protein assignment, *k*Mermaid achieves the sensitivity of popular existing protein mapping tools while remaining highly resource efficient. We evaluate *k*Mermaid on simulated data and data from human fecal samples as well as demonstrate the utility of *k*Mermaid for classifying reads originating from new, unseen proteins. *k*Mermaid allows for highly accurate, unambiguous and ultrafast metagenomic read assignment into protein clusters, with a fixed memory usage, and can easily be employed on a typical computer.

## Author summary

Whole-genome shotgun sequencing has allowed for the collection of a wealth of metagenomic data. Evidence that microbiomes play key roles in human health and disease is growing, but approaches for studying functional metagenomic content are still limited. Current protein mapping approaches do not allow for

**Data availability statement:** Availability and usage kMermaid is freely available as an open-source command line program written in Python that requires a user-provided input file containing query sequences in either fastq or fasta format. The precomputed k-mer frequency model and files containing the protein cluster members and cluster representatives are both accessible and used as internal default parameters in kMermaid. Complete download and installation instructions are found at github. com/AuslanderLab/kmermaid. Availability of data and materials All data is used in this work is publicly available. The LOTUS trial paired end sequencing metagenomic reads from ulcerative colitis patients fastq files used for benchmarking are available through SRA: PRJEB50699.

**Funding:** This work was supported by the National Institutes of Health, R01 LM014503 (to N.A.) and the V foundation V2024-006 (to N.A.). N.A. received salary from R01 LM014503 and V2024-006. The funders had no role in study design, data collection and analysis, decision to publish, or preparation of the manuscript.

**Competing interests:** The authors have declared that no competing interests exist.

direct quantification of protein coding potential because short reads commonly map to similar proteins in different bacteria. Mapping metagenomic sequencing reads to proteins in such a way that a microbiome's coding potential can be quantified is a key first step to pinpointing specific functional mechanisms or associations of disease. Here, we present a framework to first group similar proteins together, then uniquely map reads directly to these homologous protein groups. Our results show that by using *k*-mer frequencies stored in a two-layer hash map, we can sensitively classify metagenomic reads from high-depth sequencing data in only a few hours. We present our protein mapping method in an easy-to-use, resource efficient Python package, *k*Mermaid. *k*Mermaid results can be directly quantified which in turn will enable linkage of microbiome amino acid content to numerous health and disease phenotypes.

## Introduction

The gut microbiome has recently emerged as a new frontier for non-invasive biomarker discovery and new therapeutic intervention. As the field of metagenomics has matured in terms of popularity and technical advancement [1,2], there has been increasing recognition of the importance of functional analysis of microbiomes [3]. Whole-genome shotgun sequencing has allowed for the collection of vast amounts of metagenomic data which can be used to gain insights about both the taxonomic and functional composition of microbiomes. Functional profiling and quantitative comparisons of microbial proteins have immense potential to reveal microbe-microbe and host-microbe interactions, establish new microbial biomarkers, and provide predictions based on microbiomes [4,5]. However, the quantitative analyses required for these tasks are contingent on the functional classification of shotgun metagenomic reads as a preprocessing step, which remains a computational challenge.

Functional read classification is a broad notion that can encompass mapping reads directly to proteins or mapping to higher level functional classes, such as ortholog protein groups or pathways. Several methods and pipelines, such as eggNOG-mapper [6], PANNZER2 [7], BlastKOALA [8], HUMAnN3 [9](p3), and fmh-funprofiler [10], which been recently developed to perform these higher-level classifications. Such tools use a variety of computational algorithms, including alignment and sketching techniques, to report mappings to functional reference databases, such as KEGG Ortholog or eggNOG [11]. In contrast, direct protein mapping is often performed using alignment-based methods, which rely on homology between a metagenomic sequence and microbial proteins in reference databases [5]. BLASTX [12] remains the gold standard for sensitivity despite it being infeasibly slow for typical metagenomic experiments producing tens of millions of reads [13]. DIAMOND [14,15] was developed in part to address computational challenges associated with BLASTX and allows for ultrafast read-to-protein alignments, making it one of the most widely used metagenomic protein mapping tools. Another popular method, MMseqs2 [16], was primarily developed to cluster metagenomic nucleotide and protein sequences, but

also has translated protein search capabilities. While ultra-fast, neither DIAMOND nor MMseqs2 addresses BLASTX's other challenge of multimapping, i.e., when a single read aligns to more than one protein. Multimapping often occurs due to homologous proteins or domains originating in different taxa, but can be problematic for downstream read counting and subsequent analysis [17–19]. Mapping to higher-level functional classes can resolve multimapping, but not without the loss of the granular information protein annotation provides. Therefore, there is a need for resource-efficient methods that can provide unique read-to-protein level maps.

Methods for efficient and sensitive taxonomic classification of metagenomic sequences have addressed similar challenges in computational efficiency of taxonomic assignment by using *k*-mer based approaches, which are faster than alignments [20]. Most notably, Kraken [21] introduced accurate and highly efficient taxonomic classification by mapping *k*-mers to lowest common ancestors. This approach was later provided with improved resource usage through Kraken2 [22] and improved precision through KrakenUniq [23]. Other high speed taxonomic classification methods are CLARK [24], another *k*-mer based approach, as well as Centrifuge [25] and Kaiju [26], which are based on FM-indexing. Importantly, Kaiju demonstrated that the use of protein-level sequence comparisons substantially improves taxonomic classification. *k*-mer based approaches could offer similar advancements for protein mapping; however, such methods are currently lacking.

Here, we introduce *k*Mermaid, a new method for ultrafast and resource-efficient protein mapping of metagenomic reads (Fig 1, S1 Methods). *k*Mermaid uniquely maps query nucleotide sequences into taxa-agnostic clusters of highly homologous proteins using a precomputed *k*-mer frequency model (S1 Fig, Methods). The underlying rationale for *k*Mermaid is that proteins with high sequence homology have similar biological functions and thus should be grouped together for downstream analysis. To this end, mapping the sequences to clusters representing homologous groups of proteins irrespective of taxa addresses issues along both computational and biological axes. The read-to-cluster approach resolves the problem of alignment ambiguity, referred to here as multi-mapping, i.e., when a single

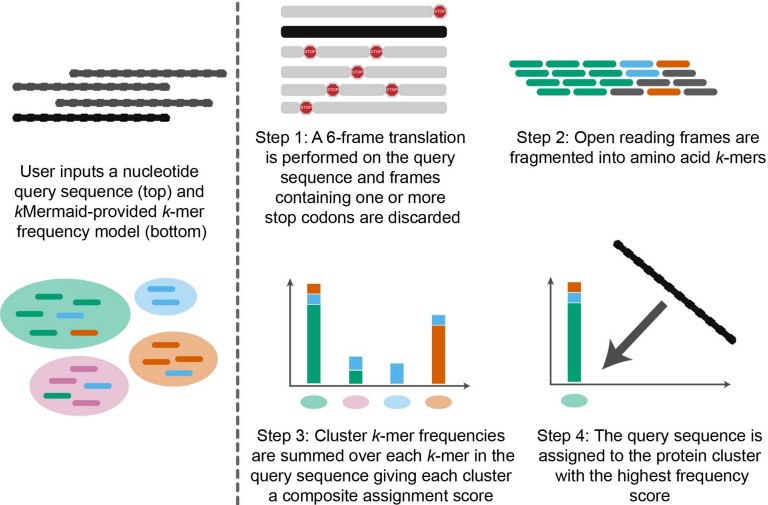

**User inputs a nucleotide query sequence (top) and *k*Mermaid-provided *k*-mer frequency model (bottom)**

**Step 1: A 6-frame translation is performed on the query sequence and frames containing one or more stop codons are discarded**

**Step 2: Open reading frames are fragmented into amino acid *k*-mers**

**Step 3: Cluster *k*-mer frequencies are summed over each *k*-mer in the query sequence giving each cluster a composite assignment score**

**Step 4: The query sequence is assigned to the protein cluster with the highest frequency score**

**Fig 1. *k*Mermaid unambiguously maps nucleotide sequences to functionally homogeneous protein clusters.** To classify a metagenomic read, a nucleotide query sequence undergoes a six-frame translation (Step 1), and frames containing stop codons are removed. Each amino acid *k*-mer in a non-truncated coding frame (Step 2) is then mapped to the protein clusters that contain that *k*-mer in the database. An assignment score is calculated to evaluate a match between the query sequence and each protein cluster, by summing frequencies of *k*-mers in the query sequence in every protein cluster (Step 3). The query is classified to the protein cluster assigned with the highest assignment score which corresponds to the cluster in which the *k*-mers of the query are most frequently observed (Step 4). A description of and pseudo-code for *k*Mermaid's implementation is provided in the S1 Methods.

read similarly aligns to multiple proteins. The resulting aligned proteins are often functionally similar but originate in different species. By aggregating at the protein cluster level, *k*Mermaid can capture novel biological effects that may be overlooked when performing analyses conditioned on taxa or when aggregating reads into broader functional categories, such as ortholog groups or pathways. *k*Mermaid can classify tens of millions of sequences in just a few hours, providing the computational speed and resource efficiency needed for the large volumes of data generated through metagenomic sequencing experiments, while matching the sensitivity of BLASTX. Through comprehensive benchmarking against other widely used metagenomic protein mapping tools, we show that *k*Mermaid achieves fast, resource-efficient, and sensitive metagenomic read classification into functional units expected to improve down-stream quantitative analysis.

## Results and discussion

### Using k-mer frequencies to map reads to homologous protein clusters

The main motivation for *k*Mermaid is that short metagenomic reads are rarely mapped to a single protein and instead often map to multiple functionally similar proteins. Obtaining a unique read-to-protein mapping requires grouping these homologous proteins into some broader functional unit. Such functional units become especially critical for downstream quantitative analyses to prevent issues such as double counting multi-mapped reads. We find that most reads map to at least five protein hits using BLASTX, which is the minimum recommended value for the number of hits reported, while only 7% of the reads can be uniquely mapped by alignment to a single protein by either BLASTX or DIAMOND in BLASTX mode (Fig 2a). In contrast, by aggregating multi-mapped hits from single proteins into homologous protein clusters (see Methods), we find that more than 93% of the reads can be uniquely mapped to a single cluster or functional unit. In other words, for more than 93% of the reads, all BLASTX hits for the read belong to a single cluster. DIAMOND follows a similar trend to BLASTX. Together, this demonstrates that our clusters resolve the majority of ambiguous alignments without loss of information from multimapping.

We developed *k*Mermaid to uniquely and efficiently map microbial short read sequencing into homologous protein clusters using this underlying clustering framework. A user can provide a file with nucleotide sequences from whole-genome shotgun sequencing to *k*Mermaid, for querying against our precomputed model of 1,793,361 proteins aggregated into 32,308 clusters. *k*Mermaid will uniquely assign each nucleotide read to a cluster and provide a readable functional annotation, i.e., protein label, based on its cluster representative. Because approximately 25% of our cluster representatives had non-descriptive names, e.g., "hypothetical protein," in RefSeq, we employed HH-suite3 [27] remote homology detection to produce descriptive cluster names. In total, we reannotated 8,617 proteins, of which 6,488 were with high confidence (S1 Table). These include 601 phage proteins, 436 membrane proteins, 230 transcriptional proteins, and 205 lipoproteins (S1 Table). The composition of *k*Mermaid's clusters is also highly consistent with preexisting, smaller-scale cluster annotations. We verified that 96% of proteins share 100% similarity with existing NCBI protein clusters [28], i.e., all proteins in the *k*Mermaid cluster also co-occur in the broader NCBI-derived clusters (Fig 2b). In addition, using keywords of NCBI-assigned protein names, we show that the functional annotations of the proteins are highly homogenous within clusters (Fig 2c). Therefore, we concluded that *k*Mermaid's cluster model and corresponding annotations are sufficiently biologically accurate and highly reflective of the cluster content.

*k*Mermaid's internal pipeline assigns sequencing reads to clusters according to a frequency-based assignment score calculated from amino acid (AA) *k*-mer frequencies. A higher assignment score indicates that *k*-mers within a query sequence are more frequently observed in the assigned cluster than other clusters in the underlying model. For some query sequences, a *k*-mer may uniquely determine cluster assignment, while other query sequences may contain multiple *k*-mers that have a higher combined frequency in the assigned (maximal) cluster compared with any other cluster. With this in mind, we reasoned that a *k*-mer found in many clusters may be less informative and its contribution to the assignment score for that cluster is noisy. In the improbable case that all *k*-mers were present in all clusters at a similar

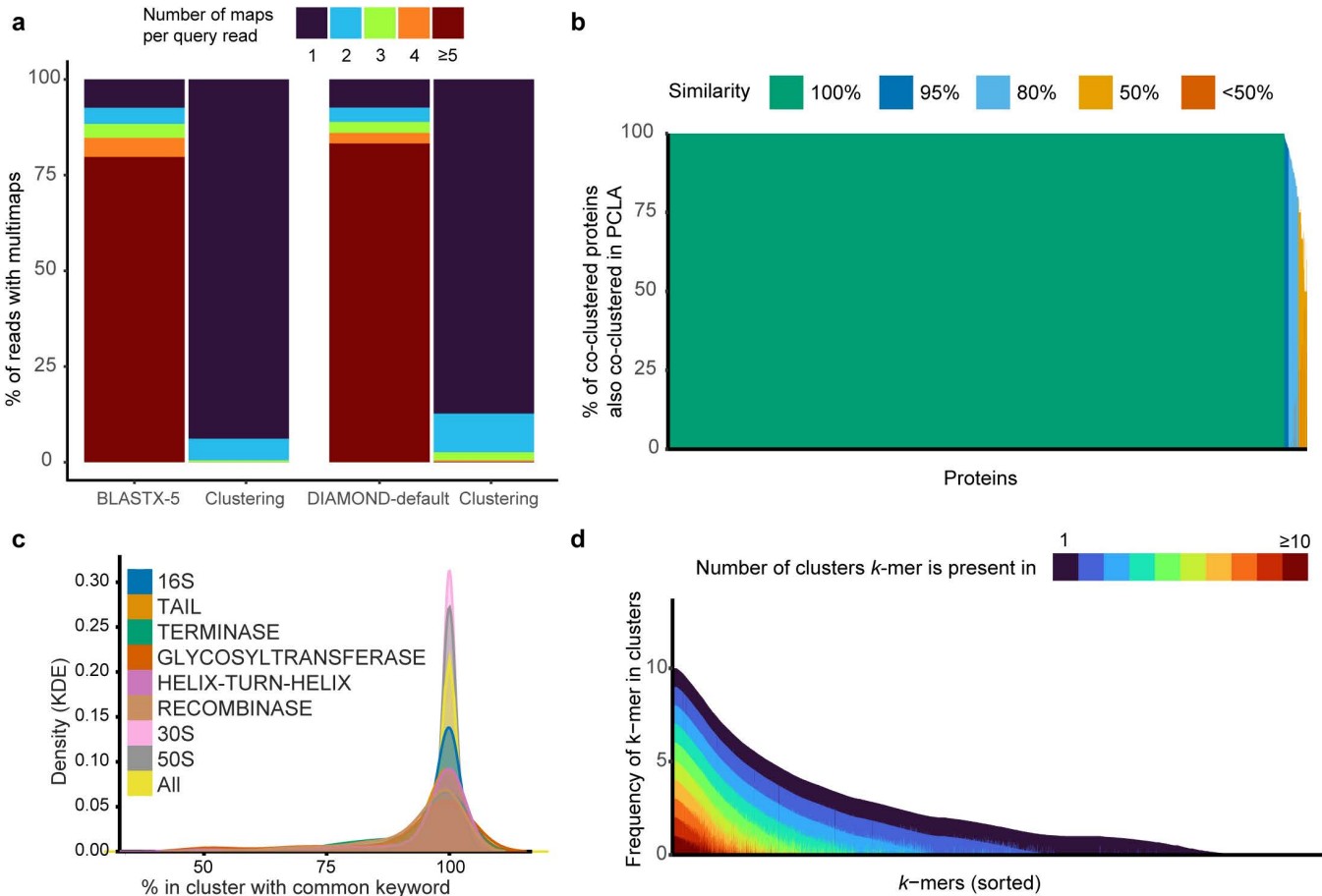

**Fig 2. Protein clusters underlying *k*Mermaid mitigate multi-mapping and allow cluster-specific prevalences of k-mers.** (a) The reduction in the number of reads mapped to >1 protein using default configurations of BLASTX and DIAMOND compared to cluster assignments, when employed to the BLASTX and DIAMOND outputs, respectively, across 29 human fecal samples. (b) The percent of co-clustered proteins using our clustering process also co-clustered in the NCBI PCLA prokaryotic protein clusters for all overlapping proteins, plotted as bars. The x-axis contains individual proteins ordered by cluster, i.e., proteins in the same cluster are plotted next to each other, and the color of the bar corresponds to the percent similarity. (c) The distribution (Kernel density estimation, KDE) of keyword percentage, i.e., percent of cluster members with the most common word from all names of proteins in the cluster, across all clusters (yellow), and for clusters with specific common keywords of interest. (d) Visual representation of the *k*Mermaid cluster frequency model sorted by the number of clusters a *k*-mer is present in. Colors represent the number of clusters in which a unique *k*-mer is found, and the y-axis corresponds to the *k*-mer frequency in the cluster. The panel shows a representative random 50K subset of all *k*-mers.

frequency, our cluster assignment would be close to random chance. On the other hand, *k*-mers that are only found in one or two clusters allow for deterministic classification and enhance our confidence in use for read assignments. Out of approximately 2.5 million AA 5-mers in our model, presence in a single cluster was the most common scenario (22%) and 81% of all AA 5-mers were found in <10 clusters (Fig 2d). We refer to these 5-mers as "deterministic," such that the presence of a deterministic *k*-mer in a query sequence is highly informative for cluster assignment. We also found that the AA 5-mers that are present in many clusters have relatively similar frequencies across the clusters when examining the top 10 clusters where they are most frequently found, implying that common 5-mers should not bias the assignment score. As anticipated, given the relative percentage of 5-mers that tend to be deterministic and that the assignment score considers multiple *k*-mers for each query sequence, *k*Mermaid clusters fully resolve read alignment ambiguity or multi-mapping in more than 95% of the cases (Fig 2a).

## Performance evaluation on simulated data with known labels

To provide validation for our model, we comprehensively benchmarked the accuracy and sensitivity of *k*Mermaid against the most widely used metagenomic protein mapping methods: BLASTX, DIAMOND, and MMseqs2. We first benchmarked *k*Mermaid using simulated data sets with varying rates of point mutations selected to mimic biological mutation rates and sequencing error rates. To compare against ground truth, the reads were simulated from the RefSeq data used in the second step of the clustering procedure (see Methods) so that their true cluster labels would be known. *k*Mermaid, DIAMOND, and MMseqs2 were almost always able to map a read to the correct protein at the level of their reporting, i.e., either the single protein (BLASTX, DIAMOND, MMseqs2) or the protein cluster (*k*Mermaid) (Fig 3a). As expected, when BLASTX and DIAMOND were restricted to reporting only one match per read, their performance dropped considerably as highly homologous proteins will have the same alignment scores. The performance of methods that align to a single protein increases substantially when viewing the singly-aligned results at the cluster level in concordance with the notion that clustering proteins by homology resolves ambiguous alignments in most cases (Fig 2a). Encouragingly, *k*Mermaid was able to assign reads to the correct cluster in nearly all cases, albeit with a slight decrease in coverage (percentage of reads classified) when query sequences are > 500 nucleotides and the mutation rates are high (Fig 3b). Because *k*Mermaid uses a cumulative scoring model, it is expected that the assignments for long, highly mutated sequences are noisier, especially at the default scoring threshold which is tuned to short reads. Similar trends were observed when we simulated reads guaranteed to have a certain number of mutations rather than using a probabilistic rate (S2a and S2b Fig). Thus, by resolving multi-mapped reads at the cluster level, *k*Mermaid shows improved sensitivity compared to methods that assign reads only to individual proteins.

## Computational efficiency of kMermaid

Shotgun metagenomics experiments commonly yield tens of millions of sequences per sample, and each must be queried against a large reference database for classification purposes. Computational efficiency remains a challenge. BLASTX is perhaps the most established nucleotide-to-protein aligner, but its computational time is infeasible for typical large files, necessitating the development of alternatives that can process these reads in a reasonable timeframe. We benchmarked *k*Mermaid's single-CPU runtime and RAM usage again against BLASTX, DIAMOND, and MMseqs2 (Fig 3c, d). DIAMOND is regarded as one the fastest accurate approaches for protein mapping of metagenomic reads and was in part developed to address the runtime limitations of BLASTX. *k*Mermaid ran 1,600 times faster than BLASTX on files with 100,000 sequences while DIAMOND ran 1,000 times faster. Both methods also provided substantial reductions in running time over MMseqs2. Both *k*Mermaid and DIAMOND were able to classify 500K sequences in a minimum of 2.7 minutes and 3.7 minutes, respectively. The same input when given to BLASTX took over six days to complete. Further, classifying 40M sequences took *k*Mermaid 3.3 hours (compared to 1.9-2.2 hours for DIAMOND), which highlights its usability for experimental shotgun metagenomic sequencing files. Like some other methods, since *k*Mermaid classifies reads independently of other reads in the same input file, it easily lends itself to parallelization by means of splitting input files into smaller chunks, allowing for further speed improvement when resources are available.

Along with speed, RAM usage is another potentially limiting factor when input files are large. We have developed *k*Mermaid to be highly memory efficient. We therefore compared *k*Mermaid to leading tools including DIAMOND, which has excellent running times, but achieves this performance in speed at the expense of higher memory and multiple CPU utilization. With these limitations in mind, *k*Mermaid performs read assignments in a way that only requires the precomputed *k*-mer frequency model, and not the input data, to be loaded into memory. As such, *k*Mermaid requires a fixed amount (2GB) of memory per run regardless of file size, whereas DIAMOND and MMseqs2 generally require memory to scale with the increasing input file size (Fig 3d). BLASTX was excluded from comparisons with more than 1 million sequences due to long running times.

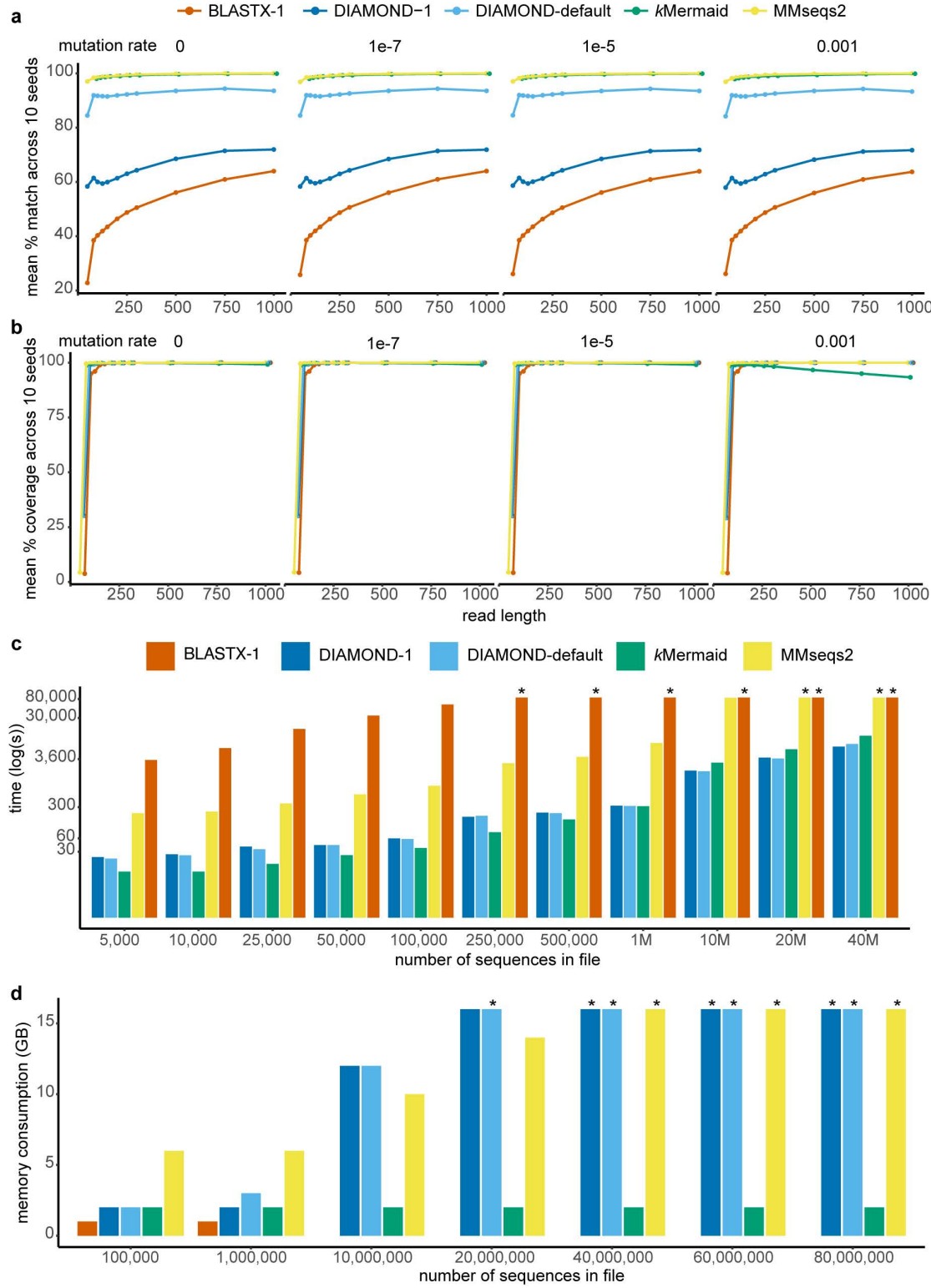

**Fig 3. kMermaid sensitivity and resource benchmarking on simulated microbial protein data.** (a) The percentage of reads classified correctly by kMermaid compared with leading read-to-protein mapping tools averaged across 10 simulated datasets per each combination of read length and mutation rate. (b) The number of reads classified by each tool normalized by the number of input reads averaged across 10 simulated datasets per

each combination of read length and mutation rate. (c) *k*Mermaid (green) provides up to a 25-fold decrease in runtimes (in seconds, log-transformed) compared to BLASTX and has comparable runtimes to DIAMOND (blue). The y-axis has been truncated and tools that exceeded a 24-hour run time for larger input sizes are denoted with an asterisk. (d) *k*Mermaid (green) requires a fixed, low memory allocation in comparison to other read-to-protein mapping tools. BLASTX was excluded from comparisons with more than 1 million sequences due to the infeasible running times. Methods exceeding 16GB of RAM are denoted with an asterisk.

## Using kMermaid to map new or unseen proteins

A key challenge in metagenomics is the classification of unknown microbial sequences that are not present in existing reference databases. To understand how *k*Mermaid performs on unknown microbial proteins, we classified segments of 22,435 new RefSeq protein sequences deposited between January and May 2025, after our frequency model was developed. We compared the resulting mappings to BLASTX alignments with the same RefSeq database used to construct *k*Mermaid's underlying database. Importantly, we observed that the *k*Mermaid assignment score is correlated with BLASTX percent identity (Spearman r = 0.83, 0.82, 0.8 for reads of length 125, 150, and 200, respectively; S3a, b Fig), highlighting the importance of choosing a more stringent score threshold when classifying reads which are likely to originate from unseen microbes. We further used the area under the receiver operating curve (AUROC) to assess the *k*Mermaid's ability to correctly classify reads, where a correct classification was based on BLASTX alignment. The *k*Mermaid was able to achieve AUROCs of 0.93 and ≥0.96 for reads matching BLASTX results and higher confidence BLASTX results filtered at a more stringent percent-identity threshold of 66.6%, respectively (Fig 4a). We observed that *k*Mermaid scores ranging from 6.3-8, depending on the read length with longer reads requiring higher scores, were able to achieve false positive rates ≤0.05 while still maintaining true positive rates around 80% (S2 Table). BLASTX was able to classify a slightly higher percentage of reads (13.7-14.9%) compared to *k*Mermaid (11.7%-13.6%), but *k*Mermaid was found to be highly concordant with BLASTX results, with around 95% of assignments correct assuming BLASTX as the gold standard (S2 Table).

## kMermaid is highly sensitive for protein cluster mapping of human fecal samples

Even though *k*Mermaid performed well on simulated reads, it is difficult to account for the additional challenges and noise associated with real, experimental data by simulations alone. Therefore, we performed additional testing using real sequencing data from 29 publicly available human fecal samples of ulcerative colitis patients [29], comparing *k*Mermaid assignments against BLASTX. On average, *k*Mermaid results agreed with BLASTX alignments 83.3% of the time and the agreement rate was highly consistent across the 29 samples (Fig 4b). Assuming BLASTX hits to be the ground truth, *k*Mermaid was able to maintain a balance between retaining a high percentage of the assignments that agree with BLASTX hits as well as a high ratio of assignments that agree with BLASTX to assignments that disagree with BLASTX (Figs 4b and S3c) at the default *k*Mermaid assignment score ≥ 3. Given that the overwhelming majority of BLASTX hits belong to a single cluster (Fig 2a), the consistent agreement between BLASTX and *k*Mermaid provides strong evidence for *k*Mermaid's ability to accurately and sensitively classify short reads from human fecal metagenomic sequencing.

## Cluster specific results

A primary objective of *k*Mermaid is to achieve high performance for classification at the read level. To comprehensively assess *k*Mermaid's performance, we also evaluated its cluster specific performance. We compared the *k*Mermaid read assignments of experimental metagenomic sequencing input samples used for benchmarking to the assignment by BLASTX, within each cluster. Interestingly, we find that clusters related to restriction, toxins, transposons, and those in the GCN5-related N-acetyltransferases family (GNAT) had high agreement with BLASTX, whereas clusters related to ABC transporters tended to have relatively low agreement with BLASTX, and therefore likely lower accuracy (Fig 4c). We also confirmed that the proportion of reads concordant with BLASTX was correlated with the mean *k*Mermaid assignment

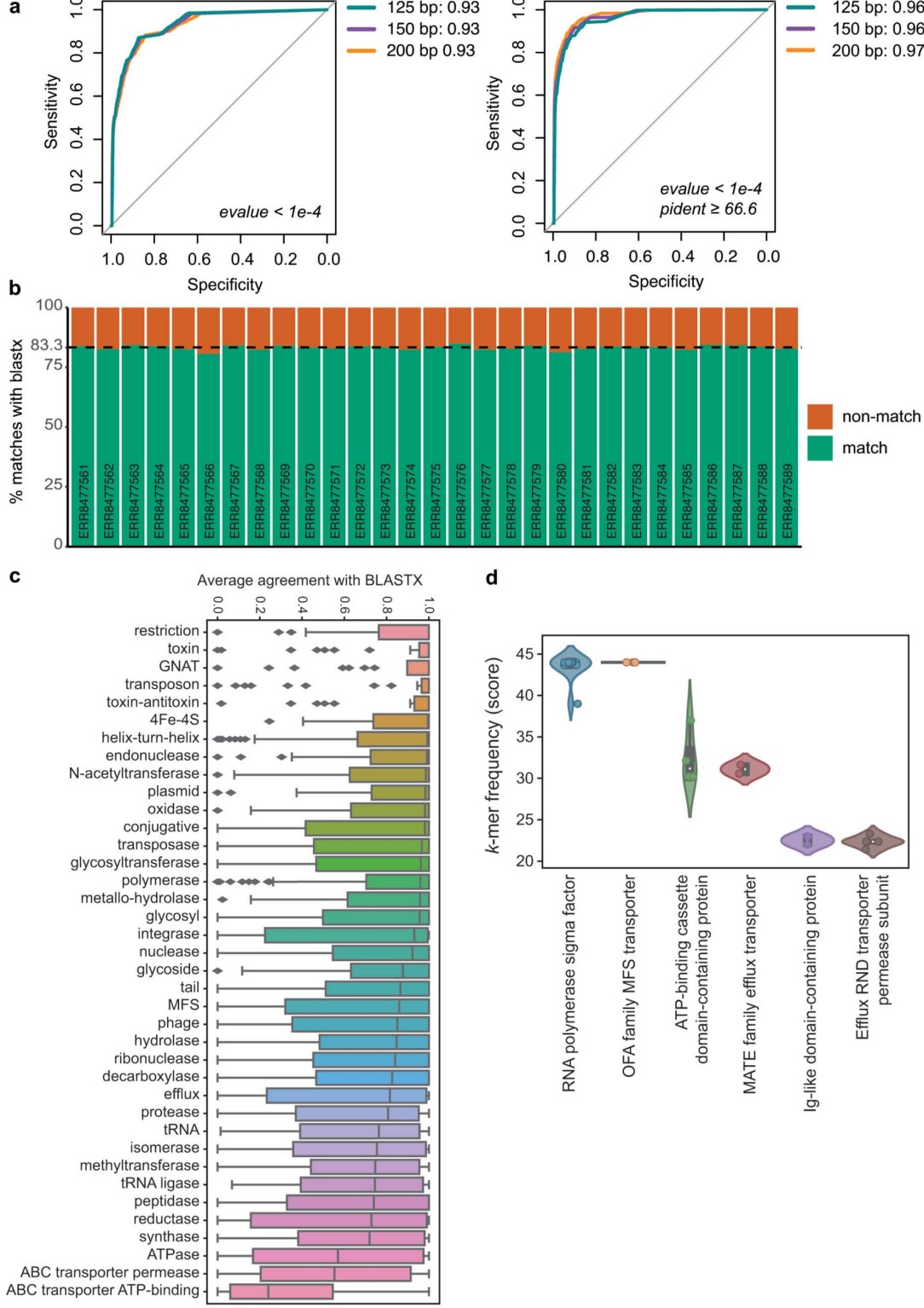

**Fig 4. Biological applications, function-specific performance, and evidence of remote homology detection of *k*Mermaid.** (a) Receiver operating curves demonstrating the ability of *k*Mermaid's assignment score to correctly classify reverse-translated nucleotide segments of varying lengths from unseen protein sequences that were added to RefSeq in early 2025. (b) Agreement of BLASTX alignments and *k*Mermaid protein assignments on 29

fecal samples from ulcerative colitis patients. (c) Boxplot showing *k*Mermaid agreement with BLASTX for clusters with specific functional annotations. (d) Violin plots showing the *k*-mer frequency scores for six clusters of reads unclassified with BLASTX that were correctly functionally classified by *k*Mermaid.

score for all reads mapping to the cluster, a trend that was not confounded by the number of reads mapped to the cluster (S3d Fig). Importantly, by investigating reads that were assigned with a high *k*Mermaid assignment score but were not classified by BLASTX, we identified reads with remote homology to proteins within *k*Mermaid clusters. We verified a correct functional classification of reads assigned to six such *k*Mermaid clusters using both PSI-BLAST [30] and HHblits3 from HH-suite3 [27] and validated these *k*Mermaid functional annotations (Fig 4d and S1 File).

## Conclusions

Metagenomic sequencing allows for the functional profiling of diverse microbes facilitating numerous biomedical applications, such as biomarker discovery and disease prediction [31–33]. To date, *k*-mer and binning based methods have been immensely useful in allowing efficient and sensitive classification of short read metagenomic sequencing into taxonomic units [21–24,34–37]. However, to the best of our knowledge, analogous methods that achieve sensitive and efficient classification at the protein level have not been developed. As a result, there remain critical limitations in our ability to classify short microbial reads for the ultimate task of downstream analysis and biological inference. Notable limitations of functional read assignment methods are ambiguous alignments and computational costs which are prohibitive for the large volumes of data typically generated by next generation sequencing experiments. The loss of granularity from higher-level functional mappings into pathways additionally prohibits analyses where amino acid sequences may be needed such as microbial peptide binding prediction [38,39]. As such, there is a need for methods that can capture and retain the underlying biology of microbial function in a granular, computationally feasible, and accessible manner.

To address these challenges, we have developed *k*Mermaid, a novel ultrafast method for the unambiguous and sensitive classification of short reads into functional units consisting of protein clusters. We show that by using a well-known concept of clustering homologous proteins into a single functional unit, *k*Mermaid rapidly resolves the majority of ambiguous BLASTX protein alignments while retaining granular amino acid level information. *k*Mermaid uses a precomputed *k*-mer frequency model based on high-confidence protein clusters encompassing almost two million microbial proteins from RefSeq. Our designated clustering allows the assignment of diverse proteins into clusters with mostly homogenous *k*-mers or words, enhancing the potential of the model to correctly capture distinct functions. Using both simulated short reads and sequencing data from real human fecal samples, we demonstrate that *k*Mermaid classifies reads with high accuracy and sensitivity compared to BLASTX but runs up to 2,500% faster on typical files with tens of millions of sequences. Additionally, we were able to verify *k*Mermaid's ability to correctly classify reads that were unclassified by BLASTX, for sequences sharing remote homology to proteins within *k*Mermaid clusters. This striking performance is likely achieved through *k*Mermaid's composite assignment scoring which uses information on a set of proteins in a cluster to classify each read in contrast to BLASTX, which is based on pairwise comparisons.

*k*Mermaid assigns a short read to a single protein cluster from a fixed set (database) based on a global maximum *k*-mer frequency assignment score, which implies certain limitations. First, in the case of ties it will randomly assign the sequence to a cluster based on the first time it reaches the maximum. Despite this, we have shown that ties should not be a widespread issue since most multi-mapped reads are mapped to a single cluster (Fig 2a) and many amino acid 5-mers are unique (Fig 2d). Second, it is possible that there exist additional clusters of proteins which are not homologous or functionally similar to any provided through our precomputed *k*Mermaid database. Because of this, it is recommended that researchers use our method as a first pass means of dealing with the computational burden of BLASTX and perform alignment-based verification for select proteins of interest. Orthogonally, additional *k*Mermaid databases/models can be (re)trained to reflect periodic increases in the quantity and diversity of available sequences. Third, although we did find

evidence supporting some ability for remote homology detection, like any method that relies on sequence comparisons, *k*Mermaid cannot classify truly novel or unseen proteins. Last, there are several biological limitations of inferring function from short reads that *k*Mermaid does not address, including misclassification of multi-domain proteins and operons. These limitations will remain for any metagenomic protein mapping tool designed for short reads and researchers wanting a more thorough view of the functional content of metagenomes should consider methods that include some degree of contig assembly prior to classification [40–42].

In summary, we present *k*Mermaid, a novel, sensitive, and runtime and memory efficient approach for the task of assigning protein identities to short microbial sequences. Future studies can utilize *k*Mermaid for the discovery of microbial functional biomarkers and as a precursor to downstream quantitative functional analyses.

## Methods

### Forming functionally similar microbial protein clusters using a two-step clustering procedure

*k*Mermaid is foremost designed to address ambiguity in read alignment, where most shotgun metagenomic reads align against multiple microbial proteins with similar alignment scores and are therefore classified as multiple functionally related proteins by the alignment (Fig 2a). To circumvent this issue, *k*Mermaid uses a well-defined concept and groups functionally related proteins into functional units prior to the assignment such that a read can be uniquely classified into a single functional unit. We therefore constructed a set of comprehensive and high confidence clusters of microbial proteins by employing a two-step clustering procedure using CD-HIT [43,44]. CD-HIT uses an incremental greedy algorithm to identify representative sequences and cluster remaining sequences by sequence similarity using short word filtering. We first clustered 43,176 proteins from the NCBI RefSeq non-redundant microbial protein database [45] using CD-HIT with a similarity threshold of 65% and a word size of 5. This first clustering step resulted in 32,308 clusters allowing non-redundant clusters. In the second step, to further expand and diversify protein members within these clusters, we applied CD-HIT-2D [44] with a 70% sequence similarity threshold to cluster all RefSeq microbial proteins against the previously selected cluster representatives (N = 1,797,426 proteins from RefSeq, dataset downloaded in May 2023). A set of expanded clusters was created from this process such that the final dataset clusters 1,793,361 proteins into 32,308 functional groups.

### Assigning functions to clusters of hypothetical or uncharacterized microbial proteins

*k*Mermaid annotates its underlying protein clusters based on the name of the representative sequence as determined in the initial CD-HIT phase. Even though these proteins are in the representative microbial protein database, 8,617 (approximately 25%) of the cluster representatives are best annotated by NCBI or RefSeq as hypothetical proteins, i.e., proteins of unknown or unverified function. To annotate these protein clusters and assign them with protein names, we used HHblits from HH-suite3 [27] for remote homology detection against two databases from The Protein Databank and UniProtKB (PD70 [46] and Uniclust30 [47] v2023, respectively) and selected the match with lowest e-value across the databases that did not map to hypothetical, unknown, or uncharacterized proteins. We were able to confidently assign a protein name or function to 6,488 of the 8,617 hypothetical clusters (e-value < 0.01) and the rest are assigned with lower confidence.

### Creating *k*Mermaid's *k*-mer frequency cluster model using nested hashing

The goal of *k*Mermaid is to functionally classify short reads using the previously defined clusters of functionally similar microbial proteins. To this end, we built a *k*-mer frequency model by obtaining all amino acid (AA) *k*-mers of all protein sequences in each of 32,308 clusters and computing the cluster-level frequency of each AA *k*-mer (S1 Fig). The value of k was chosen through hyperparameter search, by simulating truncations of each protein in the database into 50 overlapping AA segments. We evaluated k values of 3, 4, 5 and 6 for classifying truncated protein sequences, measuring accuracy as the fraction of truncated sequences which were correctly assigned to their

cluster. Both k = 5 and k = 6 achieved similarly high accuracy (>0.99), but the number of k-mers increased sharply from 2,574,615 for k = 5 to 12,043,100 for k = 6. Therefore, k = 5 was selected, achieving high accuracy with substantially fewer parameters.

We then obtained overlapping 5-mers of each protein amino acid sequence. The *k*-mer frequencies for each cluster *C* were defined by the count of the *k*-mer in the cluster *C* divided by the total number of proteins in the cluster (note that frequency can be > 1 if *k*-mers appear multiple times on average in the proteins of a cluster). The underlying model is then stored in a two-level hash map where the top-level map stores for each *k*-mer *w*, a map of clusters containing *w*, and the second level maps the clusters *C* containing *w*, to the frequency of *w* in *C*, *Cw* (S1 Methods, Algorithm 1). The resulting nested hash map can be written as $Model[w][C] \leftarrow frequency_{Cw}$. This precomputed model, consisting of 2,574,615 unique 5-mers (all the naturally occurring AA 5-mers in the underlying protein cluster database), is then used to determine the cluster to which a query sequence is assigned. The map is distributed along with the *kMermaid* package and is implemented as the default frequency model.

### Assigning protein maps to reads using the pre-computed *k*-mer model

To assign a read into a protein cluster, a six-frame translation is applied to a query sequence *r* and translations containing a stop codon (truncated frames) are discarded. Next, all overlapping AA 5-mers are extracted from the non-truncated frames. A score representing the strength of a match between *r* and each cluster *C* is then calculated by the summation of the precomputed model *k*-mer frequencies for each *k*-mer *w* in the query sequence. The score $s_{r,C}$ for query sequence *r* and each cluster *C* is computed as:

$$s_{r,C} = \sum_{w \in r} frequency_{Cw}$$

where $frequency_{Cw}$ is the model frequency of each *k*-mer *w* from *r* in cluster *C*, i.e., the average occurrence of *w* in proteins of *C*. Finally, the sequence is then assigned to the cluster with the global maximum score across all clusters (Fig 1; S1 Methods, Algorithm 2). *kMermaid* annotates the query sequence by the cluster representative for the cluster corresponding to this maximum assignment score. This scoring approach effectively assigns higher confidence to scores when *k*-mers within a query sequence are more frequently observed in a cluster. *kMermaid* uses a *k* of 5 AA chosen via hyperparameter search, as described previously, for both the base model construction and the assignment procedure described previously and reports assignments with an assignment score >3.

### Evaluating the clusters of functionally similar microbial proteins

As the protein clusters lie at the base of *kMermaid*'s approach, we validated their correctness using two orthogonal analyses aimed at verifying that the clusters produced contain homologous proteins with shared biological function.

1. *Compatibility with NCBI protein clusters.* To demonstrate *kMermaid*'s ability to construct biologically relevant clusters, we compared the results of the two-step CD-HIT clustering to the datasets from the NCBI Protein Clusters [28], which groups together proteins by sequence similarity. A subset of 102,380 of the proteins contained in our expanded cluster model was also clustered through the prokaryotic PCLA protein clusters dataset within this database. Proteins that were in this overlapping subset and were also in a non-singleton *kMermaid* cluster were used for comparison (N = 102,367, mapped to 9,984 *kMermaid* clusters). For each of these proteins, we evaluated their tendency to be co-clustered with the same proteins in both PCLA and *kMermaid* clusters by computing the percent of co-clustered proteins by *kMermaid* clustering that were also co-clustered in PCLA. The number of *kMermaid* clusters was chosen as the denominator for this evaluation metric to verify the correctness of *kMermaid* clusters rather than to assess its ability to maximize clusters, which is not an objective of this approach.

2. *Within-cluster keyword similarity.* High-throughput text analysis was performed on the protein name annotations to further investigate the similarity and homogeneity of the clusters. Trends and frequencies of word presence in clusters were used to evaluate cluster functional homogeny. After removing ubiquitous and generic words (e.g., "bacteria" or "protein"), we computed the frequency of the most common keyword found in each cluster, i.e., the fraction of proteins in the cluster containing the most common keyword in that cluster. Most clusters demonstrated a common keyword frequency of 1, indicating that our clusters are highly homogenous in key functions.

### Benchmarking against established protein mapping tools

We benchmarked *k*Mermaid against popular methods that can be used for protein mapping—BLASTX, DIAMOND, and MMseqs2. Each method was run against an underlying database containing the same 1,797,426 RefSeq protein sequences described previously. For consistency, we used the default or recommended configurations of each method. BLASTX and DIAMOND in BLASTX mode were run with e-values of 1e-4 and 1e-3 (default), respectively. MMseqs2 was run with a min-length set to 16 and e-value set to 1e-4. Each method except for BLASTX was set to report the default number of matches based on the e-value. BLASTX was set to report a maximum of 1 match for simulated data and 5 matches for experimental data due to its excessive running time when reporting all matches. DIAMOND was additionally run with a maximum of 1 match to demonstrate the consistency of mapping when using higher level groups rather than exact matches as well as to provide a fairer comparison resource benchmarking.

**Performance assessment using reads simulated from RefSeq sequences.** To demonstrate that *k*Mermaid correctly assigns proteins to clusters, we benchmarked *k*Mermaid using data simulated from nucleotide sequences of 1,383 microbial coding frames downloaded from RefSeq for which the true cluster identity is known, i.e., proteins that already exist in *k*Mermaid's model and thus have a ground-truth cluster label. From these, simulated data were generated with varying mutation rates and read lengths. Mutation rates were chosen to be representative of bacterial mutation rates [48] and error rates in next-generation sequencing data [49]. For continuous rates, the number of mutations per sequence was determined probabilistically using a binomial distribution and the location of the mutation in the sequence was determined by random sampling. A mutation was defined as a random assignment of any nucleotide that did not match the original position. Query sequences were then created by segmenting the mutated sequence to the specified read length, $l$, starting at some random position $x$ such that $x + l \leq length(read)$. Since low mutation rates could probabilistically result in no mutations, we additionally simulated reads guaranteed to have 1, 2, 3, or 4 mutations resulting in an amino acid change. To guarantee an amino acid change we employed the following procedure: 1) randomly select a substring of read length, $l$, starting at some random position $x$ such that $x + l \leq length(read)$, 2) perform a translation on the protein nucleotide sequence for the correct open reading frame based on the original sequence, 3) randomly select an amino acid to change, 4) concatenate the original substring with a randomly-selected reverse translation of the amino acid. Protein maps were assigned using the same reference database of 1,797,426 proteins that were used to develop the *k*Mermaid database using default configurations of BLASTX, DIAMOND, *k*Mermaid, and MMseqs2 or as described above. In lieu of the defaults, BLASTX was run with the recommended minimum value of 1 maximum match to accommodate a reasonable running time. DIAMOND was additionally run with only (at most) 1 match as a point of comparison for unambiguous reporting. Results from the simulations were averaged across 10 replicate datasets generated with a different random seed for each combination of parameters.

**Benchmarking computational resource utilization.** We also compared the speed and maximum memory usage of *k*Mermaid to BLASTX, DIAMOND, and MMseqs2. DIAMOND, which is up to 20,000 times faster than BLASTX, is widely considered the fastest protein aligner that can maintain the sensitivity of BLASTX results and is included in our benchmarking as a standard for efficient resource consumption. To compare the runtime of each method, we created random subsets of a single fasta file containing nucleotide metagenomic sequences from a published immunotherapy

trial of melanoma patients [50]. For running time comparisons, we tested input files with a varying number of sequences ranging from 5,000–40 million with 10 replicates each to account for machine or algorithmic variability. BLASTX and DIAMOND sequence queries were performed against the same database that was used to create the *k*Mermaid model described above. All comparisons were run on a Linux kernel using 1 task and 1 CPU per task. Because most methods can run tens of millions of sequences in under a day, we set an upper time limit of 72 hours and denote jobs that were unable to be completed in that time frame. Since no jobs took between 24 and 72 hours, we truncated the upper limit of the y-axis for visualizations to 24 hours.

Individual metagenomic sequencing experiments can yield large volumes of data and file sizes are commonly on the order of tens of gigabytes. As such, efficient memory usage is another important factor to consider when choosing analysis tools. We compared the memory (RAM) usage between all methods for input files containing 100,000, 1 million, 10 million, 20 million, 40 million, 60 million, and 80 million reads, with the latter numbers corresponding to the total number of reads commonly generated from a standard paired-end sequencing experiment (10-40M each, combined paired end). Because our SLURM cluster was not set up to report maximum memory usage, maximum RAM usage was inferred by submitting jobs with increasing amounts of memory in 1–2GB intervals (500MB, 1GB, 2GB, 3GB, 4GB, 6GB, 8GB, 10GB, 12GB, 14GB, 16GB) until the job did not report a memory related error and was able to complete successfully. For example, the max RAM for a job that was reported as requiring 12G of memory used more than 10 but less than 12GB of RAM. We denoted jobs that could not be completed with 16GB of memory, meant to reflect the feasibility of a laptop analysis. Runtime in seconds was computed manually using the date function in Linux. BLASTX was excluded from comparisons of over 1,000,000 sequences due to its infeasibly high running time, although we acknowledge that BLASTX memory usage is generally minimal.

### Analysis of unseen RefSeq microbial protein data

Since *k*Mermaid was first implemented in 2023, we were able to test the ability of our tool to classify unknown microbial proteins by using 22,435 annotated protein sequences deposited in RefSeq between January 2025 and May 2025 (obtained May 2025). Because our method uses nucleotide reads as input, we had to perform a reverse translation of the amino acid sequences. If an amino acid reverse mapped to multiple tri-nucleotide sequences without a stop codon, a tri-nucleotide sequence was selected at random, which created an even more challenging classification task. We also removed sequences with missing or non-standard amino acid sequences at this stage. We then created datasets containing randomly selected 125, 150, and 200 base pair substrings of the reverse translated sequences with 10 different seeds each. We then ran BLASTX with max_target_seqs = 5 and evalue = 0.0001 and *k*Mermaid with default configurations. Spearman correlation was computed on all BLASTX and *k*Mermaid results; datapoints were down sampled to 150,000 across all 30 datasets for visualization (S2a Fig). We additionally performed post-hoc filtering of BLASTX results using percent identity ≥66.6 to obtain higher confidence hits for comparison with *k*Mermaid. Because the reference databases for each method did not contain the truth assignment, we considered a correct hit, or true positive, to be a sequence for which the BLASTX and *k*Mermaid protein map matched. We computed AUROC using the *k*Mermaid scores for each read assigned by both BLAST and *k*Mermaid. We then the *k*Mermaid scoring threshold at which the false positive rate fell below 0.05 for each read length. These thresholds were used for subsequent comparisons with BLASTX (S2 Table). We note that while *k*Mermaid assignment scores generally correlate with higher confidence for data of consistent read length, the specific scores thresholds reported are likely specific to this analysis given the cluster-specific trends and biases in recently deposited protein sequences, i.e., over-representation of specific proteins.

### Protein mapping of reads from fecal samples from ulcerative colitis patients

The goal of *k*Mermaid is to efficiently map reads to proteins, while maintaining the accuracy and sensitivity of BLASTX. We benchmarked *k*Mermaid's functional read classification against BLASTX using paired end reads from ulcerative colitis patients enrolled in the LOTUS fecal matter transplant clinical trial [29] available on the NCBI's Sequence Read Archive

(SRA). Two samples were excluded from analysis based on data incompletion (low read depth). Due to the infeasibly long running times incurred by BLASTX, we randomly subsampled each fasta file to 100,000 reads. We used a small, representative subset (n = 3) of the available samples for in-depth follow-up analyses of all reads in the samples. BLASTX was set to max_target_seqs = 5, evalue = 1e-4 and to max_target_seqs = 3, evalue = 0.01 for these analyses, respectively. We additionally filtered BLASTX results at percent identity >66.6% where specified. We compared the overall coverage, defined as the percentage of reads that were able to be classified by both BLASTX and *k*Mermaid, as well as the percentage of BLASTX hits that *k*Mermaid was able to classify. We further investigated the correctness of *k*Mermaid's assignments using BLASTX results as a gold standard for correct read assignment, by examining reads which were assigned by both methods. Correct assignment by *k*Mermaid was defined for reads as a non-empty overlap between proteins in the BLASTX hits and the assigned *k*Mermaid clusters.

### Cluster-specific results and remote homology detection

To evaluate *k*Mermaid's performance for specific biological functions, we calculated the accuracy across clusters with similar functional annotations. We used the LOTUS clinical trial data to evaluate function specific performance in a real metagenomic sequencing cohort. To this end, for every cluster we calculated the ratio of reads correctly assigned to that cluster (i.e., assigned to that cluster by BLASTX and *k*Mermaid), out of all the reads assigned to that cluster by *k*Mermaid. Then, we evaluated the distribution of cluster performances for distinct functions, i.e., clusters named with common keywords (S3 Table).

To evaluate *k*Mermaid's ability to identify sequences with remote homology to proteins in the database, we examined reads that were not classified by BLASTX from the LOTUS clinical trial data and were assigned with high *k*Mermaid assignment scores (>20). We explored reads that were confidently assigned to six clusters that failed to be classified with BLASTX, and carefully verified that these reads have remote homology to their *k*Mermaid assigned clusters using PSI-BLAST and HHblits3 from HH-suite3 [27] (S1 File).

## Supporting information

**S1 Fig.  *k*Mermaid's internal *k*-mer frequency model with all steps outlined.**
(TIF)

**S2 Fig.  Performance on simulated reads with known labels, with 1–4 introduced mutations per read.** (a) Percent correct labels by each method for 3 typical read lengths evaluated. (b) Percent of input reads mapped by each method for 3 typical read lengths evaluated.
(TIF)

**S3 Fig.  Biological applications.** (a) Spearman correlation between *k*Mermaid's assignment score and the maximum BLASTX percent identity per read for all reverse-translated nucleotide segments (lengths = 125, 150, 200 base pairs) of each unseen RefSeq protein that was mapped by both methods without thresholding. To prevent overplotting, reads were down sampled to 100,000 (40% of total). (b) Histograms showing the *k*Mermaid score of all reverse-translated nucleotide segments (lengths = 125, 150, 200 base pairs) of each unseen RefSeq protein that was mapped by both kMermaid and BLASTX at > 66.6 percent identity. The dashed lines denote the read length-specific thresholds determined by maintaining a false positive rate < 0.05. (c) The percent of all input reads able to be classified by *k*Mermaid compared to BLASTX for sequencing from 3 representative colitis samples, chosen randomly. *k*Mermaid's optimal scoring threshold was determined by maximizing the percentage of the assignments that agree with BLASTX hits (Sensitivity, dark blue) while retaining a high ratio of assignments that agree with BLASTX to assignments that disagree with BLASTX (Accuracy, light blue). (d) Correlation between the proportion of reads concordant with BLASTX and the mean assignment scores (log-transformed) for all proteins in the cluster. Distributions of these metrics broken down by number of reads mapped

to the cluster where clusters in the bottom tertile have the lowest number of mapped reads and clusters in the top tertile contain the highest.
(TIF)

**S1 Table.  Protein names and descriptions of cluster representatives of the protein clusters underlying the *k*Mermaid model.**
(CSV)

**S2 Table.  Performance evaluation for sequences from unseen RefSeq microbial protein data.**
(XLSX)

**S3 Table.  The median *k*Mermaid model performance when classifying protein cluster containing different, specific keywords.**
(CSV)

**S1 File.  Selected reads failed to be classified with BLASTX that were correctly classified with *k*Mermaid share remote sequence homology with proteins in their associated clusters.**
(TXT)

**S1 Methods.  Pseudo-code for model training and read assignment.**
(PDF)

## Author contributions

**Conceptualization:** Noam Auslander.

**Data curation:** Jayamanna Wickramasinghe.

**Software:** Anastasia Lucas, Daniel E Schäffer, Noam Auslander.

**Supervision:** Noam Auslander.

**Writing – original draft:** Anastasia Lucas, Noam Auslander.

**Writing – review & editing:** Anastasia Lucas, Daniel E Schäffer, Jayamanna Wickramasinghe, Noam Auslander.

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
