## [Decision Letter · Decision Letter 0]

15 Apr 2025

PCOMPBIOL-D-25-00398

kMermaid: Ultrafast functional classification of microbial reads

PLOS Computational Biology

Dear Dr. Auslander,

Thank you for submitting your manuscript to PLOS Computational Biology. After careful consideration, we feel that it has merit but does not fully meet PLOS Computational Biology's publication criteria as it currently stands. Therefore, we invite you to submit a revised version of the manuscript that addresses the points raised during the review process.

Please submit your revised manuscript within 60 days Jun 15 2025 11:59PM. If you will need more time than this to complete your revisions, please reply to this message or contact the journal office at ploscompbiol@plos.org. Please include the following items when submitting your revised manuscript:

We look forward to receiving your revised manuscript.

Kind regards,

Sarath Chandra Janga, Ph.D

Academic Editor

PLOS Computational Biology

James Faeder

Section Editor

PLOS Computational Biology

**Additional Editor Comments :**

Although reviewers agreed on the simplicity and utility of the proposed metagenomic tool kMermaid, several critical concerns were raised by all three reviewers regarding the methodology, novelty, benchmarking, and clarity. In particular, I am willing to consider a significantly revised version of the manuscript which addresses the major concerns raised by the reviewers including those highlighted below.

• Improve the benchmarking of the tool by comparing speed/accuracy against MMseq2, DIAMOND and other existing tools using standardized datasets.

• Clearly make the case for the novelty of the tool by differentiating kMermaid’s approach from existing tools (e.g., cluster definitions, ambiguity metrics).

• Provide details of the method by formalizing the algorithm, revise figures for clarity, and define terms such as "truncated reads" to avoid ambiguity.

• Justify the use of protein clusters as functional proxies or adopt orthology databases. For instance, established orthology databases (KEGG, EggNOG, Pfam) are preferred for functional annotation since clustering at 65–70% identity is a poor proxy for function.

**Journal Requirements:**

6) Please amend the description of the manuscript in the online submission form to "Manuscript" rather than "Cover Letter."

**Reviewers' comments:**

Reviewer's Responses to Questions

Reviewer #1: This work describes a new metagenomic functional profiler tool. The key idea seems to be to first cluster proteins into clusters. However, this is a common approach used by all functional profilers in that they work with existing clusters of orthologous genes (e.g. KEGG OC, EggNOG, COG). Once the clusters are formed, kMermaid seems to use a very standard approach of matching kmers. The approach seems pretty straightforward and perhaps there-in lies its utility.

Specific Comments

1. Why does kMermaid have to rely on its own clusters of proteins? Why cannot it use existing databases?

2. The results in figure 2 do not seem very interesting or clear to me. For example, for part a, why would you not use BLASTX results at the cluster level as well? Also part d is hard to interpret. I am not sure I understand what is being conveyed here or that it is important.

3. Similarly figure 3 also does not seem to be conveying anything interesting in terms of a result.

4. Why does the comparison with DIAMOND also not include an evaluation of sensitivity and specificity? Also, why are other recently published tools (e.g. fmh-funprofiler: https://academic.oup.com/bioinformatics/article/40/Supplement_2/ii165/7749078) not included in this comparison?

Reviewer #2: This manuscript presents kMermaid, a tool to assign functions to short metagenomic reads. The ideas are interesting and the software is well implemented, but I feel that the authors are overselling the advantages of their approach compared to existing alternatives, mostly by artificially constraining the comparison to only one family of approaches (BLASTX/DIAMOND). The benchmarking is rather simplistic as well.

MAJOR

1. When describing the advantages of kMermaid, the authors claim that it leads to fewer ambiguous mappings, but partly this is achieved by using a different definition of ambiguity so that it applies more strictly when BLASTX is being evaluated than when kMermaid is being evaluated (Fig 2a). For example, if a read is multi-mapped to proteins A1, A2, and A3; but they all share the same cluster, then the read is ambigous at the protein level, but not at the cluster level. This is not novel in the field, but it is should be differentiated what is due to a different tool vs. a different definition of ambiguity.

BLASTX is designed to be very sensitive as well (compared to other alternatives, such as DIAMOND or mapping to nucleotide databases, such as gene catalogs), thus it will map reads to more proteins.

2. I disagree that "BLASTX remains the gold standard" for querying metagenomic reads against a large database. Despite having published many papers using metagenomics data, I have never used BLASTX for this purpose. If working in a reference-based manner, I would use nucleotide-alignment (such as bwa or strobealign) to a gene catalog (either the venerable IGC if working with the human gut [https://doi.org/10.1038/nbt.2942] or a more recent one [https://doi.org/10.1038/s41586-021-04233-4]) or a catalog of genomes (including MAGs, [https://doi.org/10.1038/s41587-020-0603-3]). Alternatively, the authors could use the set of 1,793,361 sequences they considered (after dereplicating at 97% or 95% nucleotide identity). I would use eggnogmapper [https://doi.org/10.1093/molbev/msab293] to assign functions to the genes. At the very least, the authors need to acknowledge that there are other approaches that are widely used in the field and, ideally, compare their tool to these alternatives.

3. The authors use the term "functional", but they are simply assigning to a protein cluster at 65%/70% identity. Why is this a good proxy for function? Normally, when discussing function, an orthologous group is used (KEGG being the most popular, but also eggNOG, Pfam, ... or function-specific databases such as CARD for antibiotic resistance, CAZy for carbohydrate enzymes, ...). The authors should clarify this point.

4. A typical test that is missing is to remove certain taxonomic groups from the database while using them in simulation (to mimic the case where the real world contains species/strains absent in the database). While still limited, this would be closer to a real-world scenario. I generally refrain from asking for specific experiments in reviews, but this paper is really lacking in having a more realistic benchmark.

MINOR

"human samples" -> "human gut samples"

Reviewer #3: The manuscript introduces kmermaid, which is capable of performing functional classification of metagenomes. It focuses on speed, low memory consumption and disambiguation when assigning metagenomic reads against a pre-calculated database of protein clusters.

It avoids the effort of taxonomic disambiguation by performing clustering of reads translated to their amino acid sequences and counting k-mer matches against precomputed protein clusters. The algorithm is rather simple: a single read (not assembled) is translated into all possible reading frames, and the contained k-mers are compared to a set of precomputed protein clusters in what looks like a rather brute-force approach (no indexing/hashing/sorting is mentioned). The cluster with most exact matches is chosen for functional annotation transfer.

The paper is well written and the code it is accompanied with is well structured and organized. However there are a number of major concerns I have with the manuscript.

MMseq2 [1], DIAMOND, Super-FOCUS and FMAP (Fast Metagenomic Functional Profiling) also address the same research question and are established methods (several years old). The authors should carve out their novelty with respect to them and/or ideally quantify performance measures such as speed and accuracy (the current version only contains runtime comparison to DIAMOND). The introduction does not comprehensively summarize the state of the art (except for DIAMOND). The benchmark should be designed such that a more direct and comprehensive comparison is possible. Ideally, the same benchmarks from e.g. MMseq2 could be used. Thus the benchmarking with the state of the art is not rigorous/comprehensive enough.

The main algorithm isn’t presented formally and Figure 1 lacks detail and rigor. Subfigures with labels a-e (like in some of the other figures) would be desirable, also for a more structured caption of Figure 1. Alternatively, pseudo-code could be accompanied. It is not entirely clear what is meant with ‘truncated reads’ (together with the X-shaped symbols) – it could be truncation due to instrumental limitations from sequencing technology, low quality bases, incomplete synthesis (premature termination) during sequencing or the presence of a stop codon – which one is it?

Figure 2 claims that BLASTX has a large amount of ambiguous functional maps. 1. It is not clear from the manuscript whether the BLASTX ambiguity is derived from raw scores or E-values (or something else altogether), as it seems that raw scores would also lead to less ambiguous maps. 2. How about the other state of the art methods? Do they also exhibit such high levels of ambiguity?

Fig 2a) how are the 3 metagenomic samples chosen?

2c) It seems odd that the x-axis exceeds 100% (probably an artefact from KDE)

2d) the x-axis has k-mers “sorted”. It would be best to make clear that the sorting isbased on the number of clusters, a k-mer appears in. When the text refers to Figure 2d) it mentions a single cluster being common (22%), this is not obvious from the Figure, but it could be marked. Likewise the 19% of k-mers falling into >10 clusters

How did the authors come to 2.5M k-mers. There are approx. 205 =3.2 possibilities for 5-mers. In general the choice of k is not fully motivated

The method section does not motivate some of the design choices made:

• Why 65% (1. Step) and 70% (2. Step) were used as thresholds for clustering of the precomputed model

• Why are there 2 steps, with first some 43K proteins (how are they exactly selected? From NCBI’s Prokaryotic Reference Genomes?) and then with 1.7M?

Since the 2-step clustering is non-standard and differs from plain CD-HIT, it would be good to get a sense of clustering quality as measured by a clustering coefficient like the Silhouette or Dunn index or homogeneity/separation ratio.

Was the hyperparameter (k) evaluation done according to best practices? I.e., was it conducted with proper dataset splits (inner validation sets for model/hyperparameter selection and “outer” final test set(s), which was entirely unused(!) during hyperparameter selection? How sensitive are the results to a change of k?

Limitations should be clearer. Avoiding assembly is obiouslybeneficial from a computational point of view, but lacks the uniqueness/predictive power stemming from assembled contigs.It would be desirable, if that could be quantified. The selection of 43K reference proteins most likely contains multi-domain proteins. There are occasions when read assembly is important:

1. Multidomain Proteins & Functional Context:

o Many functional proteins, especially in signaling (e.g., two-component systems) and metabolism, derive their activity from domain interactions.

o If sequence reads only cover individual domains, assigning function may be incomplete or misleading (e.g., distinguishing a full enzymatic complex from fragments).

o Assembling longer contigs helps reconstruct the true architecture of multidomain proteins.

2. Pathway Reconstruction & Gene Clustering:

o Some pathways rely on synteny (gene order and co-localization), which is lost in fragmented reads.

o Functional units like operons in prokaryotes may be misclassified if only single reads are used.

3. Taxonomic and Functional Coupling:

o If a gene is part of a mobile genetic element (e.g., plasmid, transposon), assembly can clarify if it belongs to a specific species or is horizontally transferred.

Another limitation is that beyond the precomputed protein clusters, novel proteins in metagenomes will be ignored. While this is common to many methods, it should probably be mentioned.

It would be very useful if the presented work could assess the amount of misclassification coming from unassembled reads (vs assembled reads).

Assessing the quality of the initial clustering is also important:

Regarding memory consumption, DIAMONDs strategy seems a realistic approach (scaling with input, capping at 16GB). There should be a stronger motivation for a limitation to 2GB, an amount that is exceeded by nearly any reasonable machine in use these days. It is not clear how the 2GB memory requirement is achieved – streaming/generators?

A central point is that existing methods (particularly BLASTX, not so sure about the other abovementioned methods) have a high percentage of ambiguous mappings (Fig. 2). Have the authors

Regarding speed comparisons, indexing is an extremely common method in databases to perform fast lookup. It is also used in [1]. kMermaid does not use indexing or other fast information retrieval methods hashing/LSH, sorting etc. It would be helpful to provide insides as to why such common/best practices were not deployed/necessary.

There are a few technical issues with the installation. Both on Ubuntu 20.04 (as recommended) and Mac OS 15.3, large file installation did not seem to work as described in the installation instructions: even with git-lfs installed, the file kmermaid/db/kmer_model.pkl is only 134bytes and thus not useful (not readable with pickle). The alternative method using the wget command (as per the github install instructions) does not give a larger file.

[1] Steinegger, M., Söding, J. MMseqs2 enables sensitive protein sequence searching for the analysis of massive data sets. Nat Biotechnol 35, 1026–1028 (2017). https://doi.org/10.1038/nbt.3988

**Have the authors made all data and (if applicable) computational code underlying the findings in their manuscript fully available?**

Reviewer #1: Yes

Reviewer #2: Yes

Reviewer #3: Yes

PLOS authors have the option to publish the peer review history of their article (what does this mean? ). If published, this will include your full peer review and any attached files.

**Do you want your identity to be public for this peer review?** For information about this choice, including consent withdrawal, please see our Privacy Policy .

Reviewer #1: No

Reviewer #2: No

Reviewer #3: **Yes: ** Andreas Henschel

**Figure resubmission:**
---

## [Decision Letter · Decision Letter 1]

27 Jul 2025

PCOMPBIOL-D-25-00398R1

kMermaid: Ultrafast metagenomic read assignment to protein clusters by hashing of amino-acid k-mer frequencies

PLOS Computational Biology

Dear Dr. Auslander,

Thank you for submitting your manuscript to PLOS Computational Biology. After careful consideration, we feel that it has merit but does not fully meet PLOS Computational Biology's publication criteria as it currently stands. Therefore, we invite you to submit a revised version of the manuscript that addresses the points raised during the review process.

Please submit your revised manuscript within 30 days Sep 26 2025 11:59PM. If you will need more time than this to complete your revisions, please reply to this message or contact the journal office at ploscompbiol@plos.org. Please include the following items when submitting your revised manuscript:

We look forward to receiving your revised manuscript.

Kind regards,

Sarath Chandra Janga, Ph.D

Academic Editor

PLOS Computational Biology

James Faeder

Section Editor

PLOS Computational Biology

**Additional Editor Comments :**

In light of the minor comments raised by two of the reviewers regarding the minor issues and typos as well as providing clarity on benchmarking details, the authors should submit a revised version of the manuscript addressing these comments.

**Journal Requirements:**

1) Please amend your detailed Financial Disclosure statement. This is published with the article. It must therefore be completed in full sentences and contain the exact wording you wish to be published.

1) State what role the funders took in the study. If the funders had no role in your study, please state: "The funders had no role in study design, data collection and analysis, decision to publish, or preparation of the manuscript.".

2) Please ensure that the Figures Files are uploaded in a correct numerical order in the online submission form.

**Reviewers' comments:**

Reviewer's Responses to Questions

Reviewer #1: I appreciate the authors’ considerable effort in improving the benchmark and the figures, and I think the authors have resolved most of my concerns.

Regarding my previous comments:

1. I agree with the authors that there is value in using the new database where proteins are clustered using AA words.

2. Thank you for your detailed clarification for the content in figure 2. There are some small typos: In Figure 2.a, “≤5” should be “≥5”, and in Figure 2.c, consider limiting the x-axis to be between 0 and 100%.

3. The benchmark in Figure 3 is clearer and shows that kMermaid is among the best performing tools in terms of sensitivity and coverage in most cases. The authors should specify how “% match” is calculated. Is it the same for all tools? If “% match” is the sensitivity at the cluster level for kMermaid, but at the protein level for the other tools, this might not be a fair comparison.

Some additional typos and minor issues that I spotted:

1. In the “Creating kMermaid’s k-mer frequency cluster model using nested hashing” section, the authors used “for each k-mer k, a map of clusters containing k”, which is confusing as k can both indicate a k-mer and the length of a k-mer. Consider changing it to “for each k-mer w”.

2. In the “Assigning protein maps to reads using the pre-computed k-mer model” section, the author claimed that the score would be the sum of k-mer frequencies for the k-mers in the query sequence, but in the formula, the frequency is summed over $k_i\in C$ (all the k-mers in the cluster C), which is inconsistent.

3. Do the “composite score” and the “confidence score” refer to the same thing?

4. I’m still a bit confused by the description of hyperparameter search: “k=5 was selected as it optimized the performance on truncated reference AA sequences”. It might be better to specify which metric they are optimizing: is it based on cluster purity, or AUROC of the downstream classification task, or something else?

5. “The cluster frequency” is better written as “The frequency of k-mers in the cluster”.

6. “The k-mer frequencies for each cluster C were defined by the count of the k-mer in the cluster C divided by the total number of proteins in the cluster.” Can this k-mer frequency be >1 if this k-mer appear multiple times in the same protein?

7. In Figure 3.c, consider using log scale for the y-axis.

Reviewer #2: This version addresses my previous concerns.

Reviewer #3: Most of the previous issues have been fully addressed. One remaining issue is the argument why Silhouette and Dunn index have not been calculate (the authors claim that it is impossible in the absence of centroids). No, it is not true that the Silhouette or Dunn index require the presence or calculation of centroids.

**Have the authors made all data and (if applicable) computational code underlying the findings in their manuscript fully available?**

Reviewer #1: Yes

Reviewer #2: Yes

Reviewer #3: Yes

PLOS authors have the option to publish the peer review history of their article (what does this mean? ). If published, this will include your full peer review and any attached files.

**Do you want your identity to be public for this peer review?** For information about this choice, including consent withdrawal, please see our Privacy Policy .

Reviewer #1: No

Reviewer #2: No

Reviewer #3: **Yes: ** Andreas Henschel

**Figure resubmission:**
---

## [Editor Report · Decision Letter 2]

26 Aug 2025

Dear Dr Auslander,

We are pleased to inform you that your manuscript 'kMermaid: Ultrafast metagenomic read assignment to protein clusters by hashing of amino-acid k-mer frequencies' has been provisionally accepted for publication in PLOS Computational Biology.

Best regards,

Sarath Chandra Janga, Ph.D

Academic Editor

PLOS Computational Biology

James Faeder

Section Editor

PLOS Computational Biology

---

## [Editor Report · Acceptance letter]

PCOMPBIOL-D-25-00398R2

*k* Mermaid: Ultrafast metagenomic read assignment to protein clusters by hashing of amino-acid *k* -mer frequencies

Dear Dr Auslander,

I am pleased to inform you that your manuscript has been formally accepted for publication in PLOS Computational Biology. Your manuscript is now with our production department and you will be notified of the publication date in due course.

With kind regards,

Zsofia Freund
